# Adiponectin System (Rescue Hormone): The Missing Link between Metabolic and Cardiovascular Diseases

**DOI:** 10.3390/pharmaceutics14071430

**Published:** 2022-07-07

**Authors:** Meneerah Abdulrahman Aljafary, Ebtesam Abdullah Al-Suhaimi

**Affiliations:** Biology Department, College of Science, Institute for Research and Medical Consultations, Imam Abdulrahman Bin Faisal University, P.O. Box 1982, Dammam 31441, Saudi Arabia; maljafary@iau.edu.sa

**Keywords:** adiponectin, hormone, receptor, diabetes, cardiovascular disease

## Abstract

The adipose tissue, regardless of its role in generating and storing energy, acts as a key player as an endocrine tissue, producing a wide scale of cytokines/hormones called adipokines. Adipokines such as leptin, resistin, visfatin and osteopontin own pro-inflammatory effects on the cardiovascular system in some cases. In contrast, some adipokines have cardioprotective and anti-inflammatory impacts including adiponectin, omentin, and apelin. One of the key adipokines is adiponectin, the abundant peptide regulating hormone that is released mainly by adipocytes and cardiomyocytes as well as by endothelial and skeletal cells. It acts through two main receptors: AdipoR1 and AdipoR2, forming the “Adiponectin system” which effectively exerts its cellular mechanisms and responses in target cells. It regulates various metabolic processes, while adiponectin is the adipocyte hormone known for its cardioprotective impact in clinical and experimental research. It is also a well-effector metabolic adipokine, since weight loss or diet restriction show a link with rises in adiponectin concentrations, which is accompanied with increasing insulin sensitivity, glucose, and lipids-regulation via adiponectin’s antioxidant, anti-inflammatory, anti-fibrotic actions. The high adiponectin level made it an attractive player in developing therapeutical treatments for metabolic syndromes and cardiovascular disease. The elevated plasma levels of adiponectin are mostly attributed to its benefits on cardio-metabolism. In some cases, adiponectin has been paradoxically accompanied with elevated risk of cardiovascular disease, so higher adiponectin concentration is a marker of poor prediction. Thus, the adiponectin system is attractive to researchers as a biomarker of heart disease advancement and a predictor of prognosis during the term of some cardiovascular diseases and its mechanical functions in Hypertension and diabetic patients. This review highlights the physiological roles of adiponectin as an anti-inflammatory and cardioprotective hormone as well as how it plays as a biomarker and potential therapeutic tool in the cardiovascular system in adult, children, and adolescents. The adiponectin system may be seen as a rescue hormone aiding in remodeling of the cardiovascular system on both cellular and molecular levels. The paradox role of adiponectin relevant to cardiovascular mortality should be taken into consideration.

## 1. Introduction

Adiponectin is a protein hormone discovered in 1995 comprised of 244 amino acids with a molecular weight of 28 kDa. It has a single-chain trimer structure which is a changeable N-terminal, a collagen domain, and a C-terminal round area similar to the immune complement C1. Three globular areas are located by the End-N, and C ends are linked by the Pro104-Tyr109 point. This single-chain trimer is concealed by a gong-formed shell. This structure is similar to the membrane construction of some proteins of the immune complement system, particularly C1q family and the three-dimensional shape of the tumor necrosis factor (TNF) family. The globular part is similar to the C1q-TNF superfamily, with difference in sequences of their amino acids [1].

Adiponectin exists in the blood as a dimer [2]; it also has a trimer form or protein molecule of large molecular weight hexamers and multimer (90, 180, and >400 kDa). The longitudinal structure of adiponectin is not commonly present in healthy status because of their thermodynamic instability [1,2]. Adiponectin has been related to many physiological functions such as immune response and inflammation, insulin sensitivity, energy regulation, and lipid metabolism [3]. Adiponectin is a hormone derived from adipocytes; it is found in many oligomeric forms in the plasma and has been found to be beneficial in cases of heart ischemia-reperfusion injury [4]. It has been reported that obesity affects glucose and lipid metabolism, in addition to being responsible for the alteration of heart structure and function. An increase in inflammatory cytokines is also caused by obesity. Adipokines are cytokines released by the adipocytes and have been found to have both anti-inflammatory and pro-inflammatory affects. Adiponectin, omentin, apelin, and secreted frizzled-related protein (Sfrp-5) possess anti-inflammatory and cardioprotective effects, while leptin, visfatin, tumor necrosis factor (TNF), resistin, osteopontin, and retinol-binding protein4 (RBP-4) have pro-inflammatory effects on the cardiovascular system, and preliminary findings link cardiovascular diseases to unhealthy metabolism [5]. Plasma concentration parameters of adiponectin and resistin measured in echocardiographic examination of coronary artery disease (CAD) patients were found to be related to cardiac remodeling and dysfunction [6]. Changes of the heart muscle, functional and morphological, have been noticed to be more significant in patients with chronic heart failure in addition to metabolic syndrome (MS) (combination of diabetes, obesity, and hypertension), than patients without MS [7]. A study of 47 polycystic ovary syndrome (PCOS) cases and 35 healthy subjects assessed the relation between an early diagnosis marker of atherosclerosis known as carotid intima-media thickness (CIMT), and an adipokine initially secreted from visceral adipose tissue known as omentin-1. It was found that an increase in CIMT was linked to cardiovascular risk in patients with PCOS, and serum omentin-1 was independently linked with age, as it was uncovered in early PCOS cases that it is acting as a protective acute-phase reactant [8]. Epicardial adipose tissue (EAT) has been noticed to regulate adipogenesis, insulin resistance, renin angiotensin-aldosterone system (RAAS), cardiac remodeling and output, by which it facilitates physiological and pathophysiological processes of coronary failure. Through paracrine, endocrine and vasocrine pathways, EAT secretes a wide range of adipokines, adrenomedullin, miRNA and adiponectin related to the progress of cardiovascular disease via intracellular and extracellular mechanisms [9]. A study on the protective effects of safflower oil against metabolic syndrome (MS), was carried out on 67 patients, given 8 g of safflower or placebo daily for 12 weeks. A substantial decrease in waist circumference was noted, in addition to a decrease in blood pressure, fasting blood sugar and insulin resistance, although adiponectin was noticed at an increase in patients taking safflower oil [10].

## 2. Physiology and Pathophysiology of Adiponectin on the Cardiovascular System

The role of adipose tissue as an endocrine organ is clearly indicated by the secretion of adiponectin and other adipokines, affecting the cardiovascular system at both cellular and molecular levels; it is now known to have anti-apoptotic, anti-inflammatory, and antioxidant roles, hence reducing the pathogenic effects of cardiovascular disease (CVD), therefore it is described as a “rescue hormone”. Adiponectin piles up in the sub-endothelium of heart arteries in humans whenever it suppresses the adhering of monocytes to endothelial cells and finally reduces the migration and proliferation of smooth muscles of blood vessels that promote atherosclerosis [11]. The circulating levels of adiponectin in addition to the expressed adiponectin in adipose tissue have been seen to be elevated in advanced CVD [12]. Accumulation of adiponectin was found in the heart after ischemic damage. Half-life of adiponectin was found to be longer in heart tissues than that of circulating plasma adiponectin [4].

Both hyperleptinemia and hypoadiponectinemia associated with obesity are important biomarkers to aid in the prediction of cardiovascular outcomes. The maintenance of proper cardiovascular functions is linked to normal levels of leptin and adiponectin. Cardiovascular disorders occur with insufficient adiponectin and leptin signaling, yet, in pathogenesis of cardiovascular disorders, elevated levels of both leptin and adiponectin are found [13]. Adiponectin is not only secreted by adipose tissue, but also by cardiomyocytes and connective tissue cells of the heart. Adiponectin is known to affect the cardiovascular system and its metabolism, as its decrease has been found to be a factor in the development of many cardiovascular diseases [14]. Furthermore, adiponectin has high plasma concentrations possessing anti-fibrotic, antioxidant and anti-inflammatory effects which regulate glucose levels and insulin sensitivity. Weight loss has been found to increase adiponectin levels, which in turn increases insulin sensitivity. Therefore, new drugs for cancer and cardiovascular diseases may use the adiponectin pathway, as they are diseases affected by insulin resistance [15]. The degree of vascular endothelial injury of patients with both diabetes mellitus (DM) and coronary heart disease, has been observed to be closely related to serum adiponectin, C-reactive protein levels, and insulin resistance [16].

In a study about family history of type 2 diabetes mellitus (FH-DM), 332 Japanese women aged 18–24 years were involved, and it was suggested that microvascular dysfunction may contribute to elevation of blood pressure, as FH-DM was linked to hypertension [17]. Mid-regional pro-adrenomedullin (MR-proADM) has vasodilatory effects but also is associated with cardiovascular mortality. The study noticed that MR-proADM was associated with (abdominal) obesity, through selected adipokines (retinol-binding protein-4, adiponectin, chemerin, and leptin), and subclinical inflammatory biomarkers (interleukin-1 receptor antagonist, interleukin-6, interleukin-22, myeloperoxidase, and high-sensitivity C-reactive protein) [18].

RBP4 is an adipocytokine involved in atherosclerosis, while the adiponectin plays a unique effect as an anti-inflammatory adipocytokine. Ref. [19] recruited 168 patients aged 88 with peripheral artery disease (PAD) to study their potential prognostic function in major adverse cardiovascular events (MACE). It was found that an increase in RBP4 and decrease in adiponectin serum concentrations are independently linked with PAD appearance. Furthermore, RBP4 is an autonomous predictor for MACE happening in PAD’s patients.

Figure 1 explains adiponectin sources and lists the physiological functions, while Figure 2 includes the general pathological roles of adiponectin.

## 3. Paradoxal Effect of Adiponectin

In general, the clinical importance of adiponectin as a biomarker is still greatly promising, particularly for metabolic and cardiovascular syndromes [20,21]. However, clinical explanations of blood adiponectin concentrations should be tested according to certain factors like patient’s history of CVD, gender [12,22], hypertension [23] age, and hemoglobin levels [24]. Basic science research has demonstrated useful effects of adiponectin molecule on glucose balance, apoptosis, ROS stress, chronically low-grade inflammation, and atherosclerosis, cardiac systolic, hypertension, etc. However, ref. [22] found that human research indicated that adiponectin is just a marker for glucose homeostasis, without immediate action related to the risk of type 2 DM and cardiovascular disorders. However, adiponectin has a paradox effect, the part played by adiponectin on the risk of mortality. Instead of the reversed relationship, there is a positive link between adiponectin and mortality average in several clinical cases including DM. The biology implied that this paradoxical effect is not clear. Without strong results, this may refer to adiponectin resistance and the confusing function of natriuretic peptides. Genetically, there is a direct function of adiponectin in increasing the risk of death. Since there is no creditable and robust evidence, additional studies are therefore required to uncover the clever roles of adiponectin on healthy cardio metabolism and its paradoxical role with death rate. For further explanation, plasma total adiponectin and high-molecular weight polymeric adiponectin are robustly positively associated with insulin sensitivity. However, ref. [25] demonstrated paradoxical hyperadiponectinemia in patients with acute insulin resistance because of a genetic defect in insulin receptors. This reveals one or both possible mechanisms: (1) key physiological function of insulin receptors in regulating production of adiponectin and (2) absence or dysfunction of insulin receptors affect adiponectin concentrations. Hyperadiponectinemia solved equally with decrease in antibodies of insulin receptors and dissolution of insulin resistance clinically. Figure 3 lists paradoxical effect of adiponectin

## 4. Adiponectin Resources and Multiple Functions

Adiponectin is produced by adipocytes, endothelial and skeletal cells. Cardiac myocytes are also able to produce adiponectin.

Adiponectin sources and evaluation of biological active forms of adiponectin systems must help to clarify how adiposity influences the cardiovascular system [23]. The adipose tissue-derived adiponectin showed an inverse relation between its concentration in plasma and many inflammatory markers such as C-reactive protein. Adiponectin relieves effects of inflammatory stimulators by modifying signal pathways in many target cells. The anti-inflammatory effect of adiponectin is a main factor of its useful actions on the cardiovascular system and metabolism diseases such as insulin resistance and atherosclerosis in the vascular endothelium [26,27], obesity, type 2 DM, and coronary artery disorders. Adiponectin regulates homeostasis of vascular system through key signaling pathways in endothelium cells and modifying inflammatory interaction in the subendothelium [23], while adiponectin is also available in vascular endothelial cells in humans. Endothelial cell and blood adiponectin were not similarly linked with metabolism and obesity; however, in older adults, both were oppositely linked with kidney functions. Elevated endothelial cell adiponectin concentrations are connected with elevated function of vascular endothelial, independent of blood adiponectin concentrations [27].

Clinical experiments have shown an association between adiponectin serum levels and the effect of the renin–angiotensin–aldosterone system that causes alterations in blood pressure. Antihypertensive medication with angiotensin II receptor blockers showed an increase in adiponectin concentration within six months. Adiponectin has a useful role against cardiac injury via modulating cardiac energy metabolism, pro-survival pathways, and suppression remodeling of hypertrophic cardiomyopathy. These effects of adiponectin are to be in part intermediated via the stimulation of pathways of 5’ AMP-activated protein kinase and cyclooxygenase-2 (COX-2), as well as reduce apoptosis of endothelial cells, in order to promote NO release, decrease TNF-α effect, and avoid atherosclerotic growth and migration of non-striated muscle cells [23]. Adiponectin is richly available on the surfaces of vascular and muscular tissues via a direct interaction with T-cadherin. Therefore, adiponectin is missing from T-cadherin-insufficient tissues. GPI-anchored adiponectin-binding protein has been identified as key factor for the protective effects of adiponectin on heart and revascularization [28]. In the early stages of hypertension, it is associated with hypoadiponectinemia. Serum adiponectin levels and index of Homeostatic Model Assessment may be beneficial markers for detecting patients with a risk of hypertension [29]. Adiponectin effects regulating metabolism may be partly intermediated by downstream actions of adiponectin signals in skeletal muscles. Skeletal muscles have been proven as a source for expression of adiponectin, suggesting circulating (Endocrine role) and paracrine and autocrine signals. Additionally, exercise links with adiponectin expression and blood levels in healthy and diseased individuals. Higher physical exercise is usually related to elevated adiponectin expression, but decreased adiponectin concentrations are ineffective in pre-diabetic, diabetic and obese individuals. While physical training recovers adiponectin in plasma and insulin sensitivity. Therefore, signaling of adiponectin function in skeletal muscles not only acts as metabolic regulator but has many roles in skeletal muscles through physical training [30].

## 5. Adiponectin and Its Cellular Receptors, AdipoRs “Adiponectin System”

At the cellular level, the hormone effect starts since it binds to its highly specific protein receptor molecule. This complex reaction has positioned mechanistic properties of the mode of action of the hormone on a definite biochemical base and allows elaboration of some receptor-dependent diagnostic and prognostic assays that are used now clinically and regularly [31]. Hormones signal action by their unique biochemical structures known by specific receptors on/into their target cells, via their models of release and their levels in the systemic circulation or paracrine and autocrine signals [32,33,34]. The hormone and its receptors form a highly specific system called a “Hormone-receptor system”.

“Adiponectin system” is the term which has been commonly acknowledged in the area of endocrinology as well as in adiponectin research areas since 2006, which was not far from the discovery of adipocytokines in 1984. The “Adiponectin system” term represents the specific, tight, and effective functions and communications between adiponectin source, the adiponectin hormone itself and its receptors on the target cells, which lead specifically to cellular responses in different specific target cells. This term is used widely and commonly in the Endocrine field as proven by many studies in prestigious books and publications [35,36,37,38,39,40,41,42,43,44,45,46]. The biologic effect of certain hormones is subordinated to its interactions with specific high-affinity receptors on the cell membrane or in the cytoplasm or nucleus of target cells. Therefore, receptors are interacted with signaling effector systems accountable for producing the required biologic response/s. Then, receptors deliver not only highly specific responses but furthermore convoy the mechanisms for stimulating the effector signals. Generally, receptors for the protein and peptide hormones (such as adiponectin receptors) interact with their hormones externally on the plasma cell, while steroid on the cytoplasm, but receptors thyroid hormone and vitamin D are present in the nuclear region. In some systems, receptors are available overabundantly, which is many times higher than needed to excite a maximal biological response. Despite such excessive receptor’s systems mainly seeming excrescent, they are designed to adjust a disagreement between low blood hormone (ligand)–concentration and a comparatively low-affinity (ligand)–receptor interaction—therefore, a high number of obtainable receptors, which means that the system is secured with enough (hormonal-receptor) units to completely stimulate downstream effector systems, although it is under the influence of sub-saturating concentrations of ligands [47].

CTRP9 (C1q/TNF-related protein 9) is an adiponectin paralog, related to the CTRP family, and it is beneficial to glucose metabolism as it acts as an adipocytokine [48]. It is a fat-derived plasma protein. The regulation of CTRP9 was analyzed, its levels of plasma were found to be reduced in myocardial ischemia-reperfusion injuring in addition to a decrease in CTRP9 expression in fat tissue, while an increase in plasma free fatty acid was noticed as well as an increase in the expression of NADPH in fat tissue. When ischemic heart was treated with CTRP9, a clear decrease in myocyte apoptosis was noticed, in response to stimulation of AMPK phosphorylation and hypoxia/reoxygenation, as an increase in phosphorylation of AMP activated protein kinase (AMPK) was also noticed as shown in Figure 4 [49].

The understanding of the regulation, secretion and signaling mechanisms of adiponectin are essential in cardiovascular health related to diabetes and other metabolic diseases to design new therapeutic strategies, targeting adiponectin’s antioxidant, antiapoptotic, antidiabetic, anti-inflammatory, antiatherogenic, and vasodilatory activities [50]. Various areas of the brain express adiponectin receptors (AdipoR1 and AdipoR2), including the cortex, hypothalamus, and hippocampus [51]. AdipoR1 & R2 in human have a crystal structure has transmembrane helices compose a closed inner cavity (the closed shape). In one variant of AdipoR1, that is perfectly active in regard to the main downstream signaling. Within the three molecules in the asymmetrical unity, two of them suppose the closed shape, while the third one takes on the open shape that has big entry in the inner cavity. Within the closed- and open-shape, some helices are deviated with their intracellular endings. This conversion of AdipoR1in its closed-open shape, may be support its physiological mechanism. Both AdipoR1 and AdipoR2 molecules have an essential anti-diabetic action [52]. 

Insulin sensitivity is regulated by AdipoR1 through the activation of the AMP-activated protein kinase (AMPK) pathway (found in the cytoplasm, AMPK phosphorylation initiates many key gene regulatory functions of AMPK in different cells [51], while inflammation and oxidative stress are inhibited by the stimulation of neural plasticity through AdipoR2 activation of the peroxisome proliferator-activated receptor alpha (PPARα) pathway [53]. PPARα is found in the cytoplasm, it regulates the expression of genes involved in fatty acid beta-oxidation, and is a major regulator of energy homeostasis [51].

Figure 5 shows the organs expressing adiponectin receptors.

Decrease in adiponectin was found to be associated to diabetes linked diseases such as cardiovascular disease and insulin resistance in type 2 DM. Endoplasmic reticulum (ER) resident protein 44 (ERp44), glucose-regulated protein 94 (GPR94), disulfide-bond A oxidoreductase-like protein (DsbA-L), and ER oxidoreductase 1-α (Ero1-α) are ER’s proteins involved in the synthesis and secretion of adiponectin higher-order complexes; these complexes have been proven to have essential action in metabolic tissues [54]. Adiponectin receptors, AdipoR1 and AdipoR2, mediate metabolic processes [15]. In hypertension, the adiponectin stimulatory effect on sodium excretion is impaired; this is credited to the increased expression and activity of G protein-coupled receptor kinase 4 (GRK4). Targeting GRK4 is a novel approach against hypertension [55]. AdipoR1 and R2 are downregulated in type 2 diabetes and obesity, causing a decrease in cardiac adiponectin signaling, reducing cardiac protection against cardiac remodeling and lipotoxicity. An adiponectin receptor-mediated signaling pathway is possibly a futuristic approach for diabetic cardiomyopathy [56]. Inflammatory response syndrome is provoked in cardiopulmonary bypass (CPB) surgery through signaling of both Toll-like receptor 4 (TLR4) and tumor necrosis factor α (TNF-α). Figure 6 illustrates multiple adiponectin receptor pathways.

## 6. Cellular Effects of Adiponectin on Neurophysiology of the Heart

The effect of adiponectin and secreted protein acidic and rich in cysteine (SPARC) on nerve growth factor NGFβ was investigated and found to have opposite regulatory functions, where PC12 cells treated with SPARC alone did not result in neurite outgrowth or activation of mitogen-activated protein kinase (MAPK). Although enhanced neurite outgrowth and NGFβ-induced MAPK phosphorylation were detected when cells were treated simultaneously with SPARC and NGFβ, cells treated with adiponectin alone increased AMP-activated protein kinase (AMPK) phosphorylation but did not induce neurite outgrowth. Simultaneously treated cells with NGFβ and adiponectin caused a decrease in the number and size of neurite cells [57]. Adiponectin has protective effects on neural stem cells and neurons [3]. In a study by [58], cardiovascular autonomic neuropathy (CAN) was measured by heart rate (HR) and heart rate variability (HRV) in participants of the Whitehall II cohort study. Where increased levels of IL-1 receptor antagonist (IL-1Ra) were seen as risk markers for the increase in HR, increased levels of adiponectin in type 2 diabetes individuals were linked to the progression of favorable cardiovascular autonomic functions [58].

## 7. Adiponectin Plays the Key Link between Metabolic and Cardiovascular Diseases in Adults

The adiponectin system plays critical functions in converging the missing links between metabolic disorders and its complications on the cardiovascular system, including the role of adiponectin itself, AdipoRs as well as the anti-inflammatory and protective effects of adiponectin on cardiac functions.

Adiponectin concentrations are lower in women vs. men, obese individuals when compared with leans, and type 2 DM vs. non-DM. In both human and experimental animals, there is a robust inverse link between plasma adiponectin concentrations, visceral fat, and body mass index (BMI). Decreased concentrations of adiponectin are positively linked with an increase in occurrence and risk of obesity, insulin resistance, DM, decreased HDL, and increased TG leading to the initiation of vascular disorders [59]. Many clinical research has showed significant reduction in concentrations of adiponectin in patients with type 2 DM since adiponectin concentrations are negatively associated with myocardial ischemic risk [60]. Pathways required in the adiponectin system to perform its actions between metabolic and cardiovascular disorders are reported by Lee and Kwak [61], as many adipokines participate in the progress of these disorders. On the contrary, adiponectin plays a curative goal for metabolic and cardiovascular disorders. Blood concentrations of adiponectin are significantly decreased in obesity, diabetes, hypertension, and coronary arteries disorder in human and experimentally induced models of insulin resistance and diabetes in animals through agonist of AdipoRs to treat the type 2 DM obese mouse model. On the other hand, adiponectin can act on its specified AdipoRs in several tissues via its different signaling mechanisms as mentioned previously—autocrine, paracrine, and endocrine. It regulates apoptosis and cell survival and growth. It exerts important pathophysiological effects in metabolic functions through its metabolic effect on glucose and lipids in liver and skeletal muscles [62].

Understanding adiponectin signaling pathways is essential for future therapies for type 2 DM and causes of insulin resistance [63]. In apolipoprotein E deficiency cases, adiponectin was recognized to have a direct effect decreasing inflammatory responses caused by mechanical injury in vascular endothelium and increasing endothelium protection through stimulation of phosphorylation and adenosine mono phosphate (AMP) kinase activation. Some dietary modifications, antidiabetic, and cardiovascular medications, and fibric acid derivatives were confirmed to increase adiponectin levels [2]. Adiponectin has a protective effect on colon, lungs, heart, and vesicles through its anti-inflammatory and anti-apoptotic properties [64]. As a result of inflammation, an increase in the concentration of circulatory cytokines is observed in patients with metabolic syndrome (MS). Different cell types secrete inflammatory cytokines into the circulation, regulating various tissues through peripheral central and local actions. Adiponectin is associated with insulin sensitivity, and it aids in the prevention of T2DM [65]. Foam cells form due to the accumulation of lipids in macrophages in atherosclerosis, and adiponectin demonstrated the ability to suppress lipid accumulation in macrophages, through the AdipoR1 and AdipoR2 and the APPL1 docking protein (APPL1 interacts directly with the intracellular region of AdipoR1 and AdipoR2 found on a cell membrane [66]). Additionally, AdipoR1 and AdipoR2 regulate scavenger receptor A type 1(SR-A1) (on cell membrane) and inflammatory cytokines. Hence, the adiponectin-AdipoR1/2-APPL1 axis can prevent the formation of macrophage foam cells preventing atherosclerosis [67]. On the other hand, it was uncovered that both the increase in adiponectin concentration and the increase in brain natriuretic peptide (BNP) increased mortality in patients with stable coronary artery heart disease (CAD), whereas the increase in adiponectin was not related to coronary revascularization, CV death, non-fatal myocardial infarction, or stroke [68]. The biology of adiponectin may vary in patients with heart failure (HF), specifically patients with preserved ejection fraction (HFpEF) (which comprises 50% of all HF), as most HFpEF patients are obese, causing the release of natriuretic peptides due to inflammation and myocardial stiffness in comparison with patients with reduced ejection fraction (HFrEF). In addition, analysis using Mendelian Randomization (MR) revealed that adiponectin levels were not associated with coronary artery disease (CAD), although the study did not take into consideration the history of CAD and coronary heart failure, which may alter the analysis outcome [69]. Due to the gradual increase in the levels of adiponectin in the progression of heart failure, there is a growing interest in adiponectin as a marker of this disease [14]. The ability of serum adiponectin levels to predict the progression of acute coronary syndrome (ACS) (conditions causing a sudden decrease in blood flow) was studied, and it was concluded that increased cardiovascular events in male patients aged ≥65 years, with BMI <25 kg/m^2^, were linked to an increase in adiponectin levels. However, the predictive capability is limited in female patients <65 years, in addition to patients with BMI ≥25 kg/m^2^ [70]. Cardiometabolic health benefits from high concentrations of adiponectin, although some studies link adiponectin to a higher risk of heart failure in individuals suffering from high cardiovascular risk. A study investigated the link between adiponectin and heart function parameters, in addition to determining heart function and N-terminal pro b-type natriuretic peptide (NT-proBNP) genetically using Mendelian randomization, as the main markers of heart failure on adiponectin. A weak link between lower diastolic parameters E/A ratio (early filling phase (E) and atrial contraction (A)) and higher adiponectin concentrations was found, but no link with the function of the left and right ventricles was found. No direct effect of adiponectin on left ventricular function was found using genetic analysis, although results did show a link between adiponectin serum concentrations and NT-proBNP [71]. With the progression of age, the risk of cardiovascular disease increases regardless of sex. This increase in risk in part may be linked to the alterations of hormones. Adiponectin is regulated by hormones, and they fluctuate throughout life. Plasma adiponectin concentrations increase in both men and women with progression of age; compared with men, pre- and post-menopausal women have higher concentrations of adiponectin. At younger ages, the protective effects of adiponectin are clear. Nevertheless, there are still risks of cardiovascular diseases despite an increase in adiponectin levels over time [72]. Adiponectin was found bound to cells that express a glycosylphosphatidylinositol-anchored cadherin known as T-cadherin but is not found bound to cells that express other receptors such as AdipoRs or calreticulin. This binding causes endocytosis and the build-up of adiponectin in multivesicular bodies increasing secretion by exosome biogenesis. An increase in regeneration of skeletal muscles and the decrease of endothelial cell ceramides was noticed with the increase in exosome biogenesis. Adiponectin in mesenchymal stem cell therapy was used in mice genetically altered with deficiency of T-cadherin or adiponectin as they were reported to have low exosome levels in plasma [73]. Myocardial ischemia/reperfusion (MI/R) was found to release small extracellular vesicles enriched with an miR-23-27-24 cluster, causing significant endocrine dysfunction and adipocyte endoplasmic reticulum (ER) stress. Using this information and targeting, these extracellular vesicles may be a potential therapeutic prevention of metabolic dysfunction after MI/R [74]. Diabetic cardiomyopathy is the association of increased heart failure in diabetic patients, increased oxidative and endoplasmic reticulum stress, increased cardiac fibrosis, pathological hypertrophy as well as diastolic dysfunction, which are characteristics of diabetic cardiomyopathy. Effective treatment is yet to be found; although in T2DM and obesity, adiponectin was reported to be a cardioprotective adipokine that is downregulated, adiponectin signaling is also impaired due to the downregulation of both receptors (AdipoR1 and R2), leading to reducing cardio protection. The protection of adiponectin signaling was found to be against cardiac remodeling and lipo-toxicity [56]. Adiponectin receptor agonists are small molecules with great therapeutic value; adiponectin mimetic compound ALY688 functioned to stimulate acetyl-CoA carboxylase (ACC) phosphorylation and AMP-activated protein kinase (AMPK) induce glucose oxidation and decrease fat oxidation in adipocytes of both epididymal (Epid) and subcutaneous inguinal (Sc Ing) adipocytes. Table 1 clarifies the effect of ALY688 on both (Epid) and (Sc Ing) [75]. Improved insulin sensitivity was performed by the ALY688 adiponectin signaling pathway activation in skeletal muscles. ALY688 greatly increased phosphorylation of AMPK, ACC, and p38MAPK [76].

## 8. Adiponectin Is a Link between Risk Factors of Cardiovascular Diseases in Children and Adolescents

Adiponectin not only has crucial roles in adults as it showed its anti-atherogenic, anti-inflammatory, and insulin-sensitivity actions. However, adiponectin plays significant functions even in children and adolescents. Studies proved lower levels of adiponectin in obesity in children and adolescents, while biomarkers of proinflammatory and inflammatory and cytokines are greater. Hypoadiponectinemia may act as a factor in the low-grade systemic chronic inflammatory case linked with obesity in children. In addition, 589 obese children and adolescents participated and were exposed to glucose tolerance assay; it was obtained that low concentrations of adiponectin are linked not only with higher C-reactive protein concentrations, but also with parameters of the metabolic disease, which includes a lipid profile. This link between adiponectin concentrations and C-reactive protein, the robust inflammatory marker, is separated from adiposity and insulin resistance in obese children and adolescents. Adiponectin could act as a member of the signal’s family connecting obesity and inflammation. Therefore, adiponectin may act as a biomarker for the metabolic syndrome in obese children [77]. Another study conducted on 632 French Canadian youth of 9, 13, and 16 years in 1999. The boys group showed 17% lower levels of adiponectin than girls. At 16 years of age, an average adiponectin level was 27.7% in boys and 13.3% in girls lower than the 9-year-old group. A mean level of adiponectin reduced relatively with every unit rise in body mass index. Male sex and alterations in fat may be main determinants of the reducing adiponectin levels of growing group of youth, which joined with a dissociation of adiponectin and insulin resistance’s parameters. The connection between obesity and parameters of insulin resistance is reduced in youth of higher adiponectin levels, which made adiponectin a potential mediation goal or risk biomarker [78]. A recent study agreed with two mentioned studies as it was reported that adiponectin is linked with risk factors for cardiovascular diseases in obese children. Plasma concentration of adiponectin is inversely linked with abdominal obesity. Additionally, decreased levels of adiponectin are related with the progress of metabolic syndrome, hypertension, and insulin resistance in pediatric individuals. With a higher number of cardiovascular risk elements, plasma adiponectin level is inversely linked with early injury of organ, such as an increase in thickness in carotid intima-media. A low adiponectin concentration in children may detect the progress of atherosclerosis in adulthood, thus adiponectin could be used to apply cardiovascular risk in obesity of children, while changes in lifestyle could lead to high plasma of adiponectin. Adiponectin could act as an early detecting marker of cardiovascular risks in children [62].

## 9. Significant Cellular Roles of Adiponectin as a Rescue Hormone In Vivo (Experimental Animals) and In Vitro Studies

Adiponectin may also be a factor in the progress of hypertension. Ref. [79] reported that, on a high-salt meal, adiponectin-deficient experimental animals show a noticeable elevated systolic blood pressure in comparison with wild-type control animals separately from insulin resistance. In genetically obese mice, overexpression of the adiponectin hormone may minimize the systolic blood pressure [26,79,80,81]. Adiponectin receptors are also expressed in the liver and the skeletal muscles. In rats with STZ- induced diabetes, an increase in the expression of AdipoR1 was reported, yet the cardiac inflammatory response was also increased, while glucose transporter 4 (GLUT4) protein was decreased linked to the reduction of circulating adiponectin [82]. Adiponectin stimulation of primary cardiac fibroblasts increased total collagen; hence, adiponectin mediates ECM remodeling after pressure overload (PO), and any deficiency in adiponectin will result in the delay of remodeling [83]. In a study on wild-type mice (adiponectin^−/−^), the animals were exposed to 30 min of myocardial ischemia and then followed two times of reperfusion within 24 h. Treatment of adiponectin 10 min before reperfusion decreased myocardial iNOS protein expression induced by ischemia/reperfusion, reduced NO/superoxide release, suppressed peroxynitrite synthesis, and inversed pro-apoptotic and infarct-expansion noticed in adiponectin^−/−^ mice. Adiponectin is a naturalistic hormone that protects the heart from ischemia/reperfusion- induced harm [60].

Adiponectin paralog CTRP1 (C1q/TNF-α (tumor necrosis factor-α)-related protein 1), was confirmed to be associated with diabetes and cardiovascular disease. In mice, the CTRP1 role in cardiac function was studied in post myocardial infarction (MI). An increase in survival rates was seen in mice with CTRP1 deficiency, as well as an improvement in cardiac function, a decrease in oxidative stress levels and inflammation, in addition to reduced infarct area. However, few differences were seen in cardiomyocyte levels of inflammation and oxidative stress, with the absence or treatment of CTRP1, in comparison with the control group under hypoxic conditions. It was also revealed that macrophage activation was regulated by CTRP1 through adiponectin receptor 1, which in turn binds to Toll-like receptor 4 (TLR4) on the macrophage membrane. Hence, cardiac function post MI is suppressed by CTRP1 by TLR4 on macrophages. Targeting CTRP1 may be used as a therapeutic management to treat cardiac dysfunction post MI [84]. In the rat model, AdipoRon (an adiponectin receptor) was found to decrease inflammation and weaken heart functions caused by CPB through mediation of AMPK that upregulates immunosuppressive IL-10 and inhibits proinflammatory cytokines like TLR4 and TNF-α in cardiac cells [85]. Mesenchymal stem cells have been used therapeutically in mice for pressure overload-led heart failure, which involved the existence of plasma adiponectin and T-cadherin expression as well as biogenesis of exosome in the treated mesenchymal stem cells with a high level of exosomes in plasma. The key role of exosome-induced by adiponectin is pleiotropic organ security [73]. Additionally, bone marrow mesenchymal stem cells (BMSCs) have been used as therapeutical potential for DM and heart disorders. For optimization, the effects of these mesenchymal stem cells on suppressing myocardial fibrosis were investigated and studied in DM models in vivo and in vitro. Although DM rats presented diabetic symptoms such as low cardiac function, expression of pathological injures and collagen, these weaknesses were markedly inhibited by the adiponectin modified mesenchymal stem cell therapy, without affecting adiponectin siRNA modified mesenchymal stem cell treated diabetic rats. Adiponectin modulated mesenchymal stem cells could decrease the expression of TGF-beta1/smad signaling pathway, to inhibit the myocardial fibrosis in the DM rats and elevated glucose stimulated H9c2 cells. Adiponectin modified mesenchymal stem cells revealed protection on cardiac fibrosis, so modulated mesenchymal stem cells may be a new therapeutic strategy for the diabetic cardiomyopathy in the future [86].

Table 2 shows the effect of adiponectin in clinical research (adults and children) and in vivo studies in different health conditions.

## 10. Adiponectin Effects on Cardiac Cells and Remodeling

Adiponectin has been found to activate the 5′ AMP-activated protein kinase signaling cascade, directly affecting cardiac cells, benefitting acute cardiac injury by aiding with cardiac remodeling [87]. Leptin and adiponectin inhibit collagens and matrix metalloproteinases, revealing the capabilities of adipokines to remodel the extracellular matrix of myocardial cells. In addition, the pathogenesis of heart failure in obese individuals may be identified by the alterations in adipokine profiles [88]. Left ventricle (LV) hypertrophy was reported not to be affected by the deletion of adiponectin, although adiponectin maintained mitochondrial oxidative power, and it prevented the remodeling of an LV chamber. It seems that, in response to pressure overload, adiponectin may have a role in alterations of cardiac metabolism and structure [89]. Telmisartan, a blood pressure medication, reduced the effects of isoproterenol induced cardiac remodeling and cell injury in male Wistar rats correlated with the increase of cardiac expression of adiponectin [90]. In a study of [83] on adiponectin knockout (AdKO) and wild-type (WT) mice, the extracellular matrix (ECM) remodeling of cardiac cells was observed in relation to pressure overload (PO). Gene expression of myocardial adiponectin was reduced after being subjected to PO for four weeks, while the levels of adiponectin in cardiac homogenates were increased in spite of the decreased levels of circulating adiponectin. The accumulation of collagen was found to increase in WT mice after PO of two and four weeks, while fibrosis was absent in AdKO mic after two weeks but abundant after four weeks. Gene array analysis is indicated in Figure 7.

Heart adaptation long-term exposure to stressful stimuli, such as ischemia/reperfusion and hypertension, eventually leads to heart failure. Adaptation is known as cardiac remodeling. Autophagy is thought to be a mechanism controlling cardiac remodeling and causing heart failure. Autophagic response was affected by leptin and adiponectin dysregulation, causing acceleration of cardiac remodeling. Elevated levels of leptin in obesity are linked to hypertension and acute cardiovascular events. Reduced levels of adiponectin in obesity are related to pathogenesis of cardiovascular diseases. Cardiac remodeling and heart failure are affected by induced changes in autophagic flux by leptin and adiponectin [91]. There are many different cells in the myocardium where endothelial fibroblasts and cardiomyocytes are the most abundant. Paracrine signaling between these cells is important for normal cardiac function, in addition to impacting cardiac remodeling and heart failure. The ligand–receptor pairs found in many cells, including the main cell types in the heart, indicates that autonomic signaling is widespread. In cardiac remodeling and heart failure, autocrine signaling is involved in hypertrophy, angiogenesis, fibrosis, inflammation, and cell survival [92]. Elevated adiponectin levels in HT and DM patients were linked to impaired mechanical functions of the left atrium. Adiponectin elevated levels may be predicted independently through age and hemoglobin levels [24].

## 11. Conclusions

Adiponectin is an important adipocytokine released by adipocytes, cardiac myocytes, endothelial, and skeletal cells; its receptors are widely distributed in brain cortex, hypothalamus, and hippocampus, as well as in the heart, liver, and skeletal muscles. AdipoR1 and AdipoR2 mediate metabolic processes and accumulation of lipids in macrophages in atherosclerosis; adiponectin demonstrated the ability to suppress lipid accumulation in macrophages. Adiponectin concentrations are accompanied with increasing insulin sensitivity, glucose, and lipids-regulation via adiponectin’s antioxidant, anti-inflammatory, and anti-fibrotic actions. These receptors also regulate scavenger receptor A type 1(SR-A1) and inflammatory cytokines. Adiponectin has a dual effect on the cardiovascular system. It has protective effects including antiapoptotic, anti-inflammatory, antioxidant, antifibrotic, vasodilatory roles, while, in some cases, it behaves as a pro-inflammatory factor such as cases accompanied with natriuretic peptide, as an increase in adiponectin concentration and the increase in brain natriuretic peptide was found to increase mortality in patients with stable coronary artery heart disease. Additionally, adiponectin has protective effects on neural stem cells and neurons that relate to the heart. Elevated levels of leptin in obesity are linked to hypertension and acute cardiovascular events. Reduced levels of adiponectin in obesity are related with pathogenesis of cardiovascular diseases. This adipocytokine plays a role in activating 5′ AMP-activated protein kinase signaling cascade, directly affecting cardiac cells, benefitting acute cardiac injury by aiding in cardiac remodeling. Therefore, adiponectin is a potential tool to be used as a predictor and therapeutic management for cardiac disease considering sex, age, and patient history in adults as well as in children and adolescents.

## Figures and Tables

**Figure 1 pharmaceutics-14-01430-f001:**
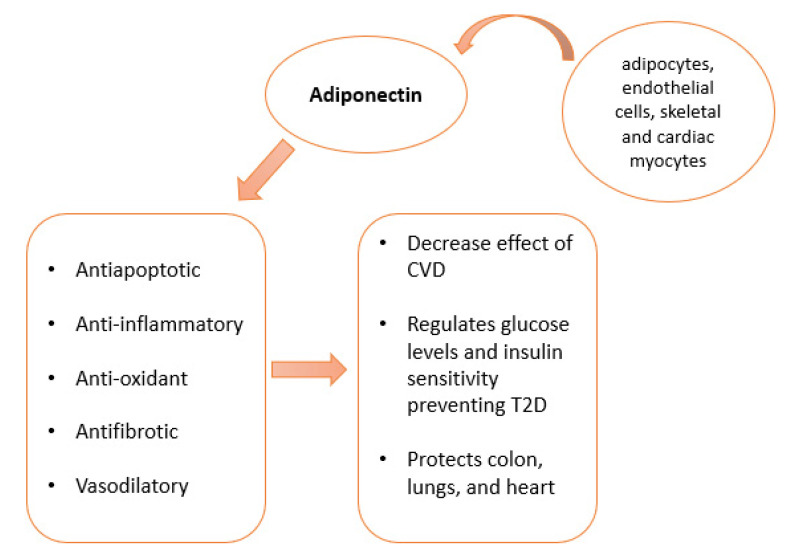
Adiponectin sources: Adiponectin is produced by adipocytes, endothelial and skeletal cells. Cardiac myocytes are also able to produce adiponectin. Its functions include antiapoptotic, anti-inflammatory, antioxidant, and anti-fibrotic effects. Adiponectin protects against CDV, controls glucose and insulin, and protects against heart, lung, and colon diseases.

**Figure 2 pharmaceutics-14-01430-f002:**
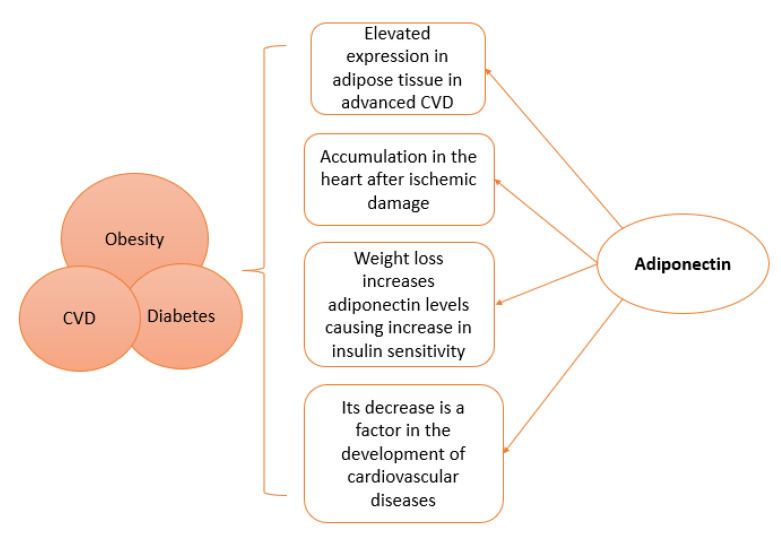
Physiological and pathological links of adiponectin to Obesity, Cardiovascular disease (CVD), and Diabetes.

**Figure 3 pharmaceutics-14-01430-f003:**
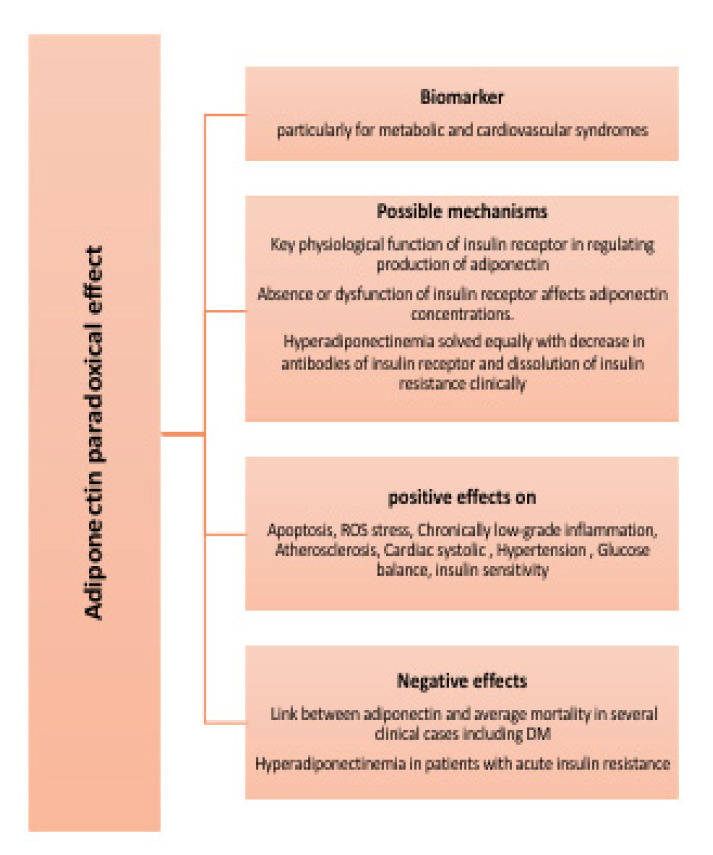
Paradoxical effect of adiponectin.

**Figure 4 pharmaceutics-14-01430-f004:**
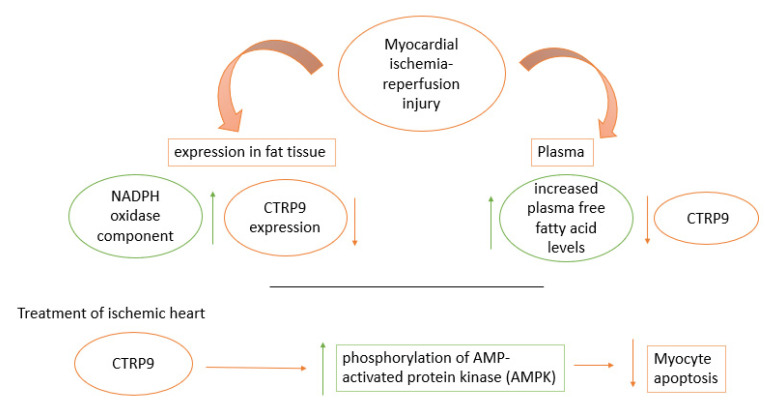
An increase in CTRP9, an adiponectin paralog, was found to increase in myocardial ischemia- reperfusion injury in plasma and fat tissue expression. When ischemic heart was treated with CTRP9, an increase in phosphorylation of AMPK led to a decrease in myocyte apoptosis.

**Figure 5 pharmaceutics-14-01430-f005:**
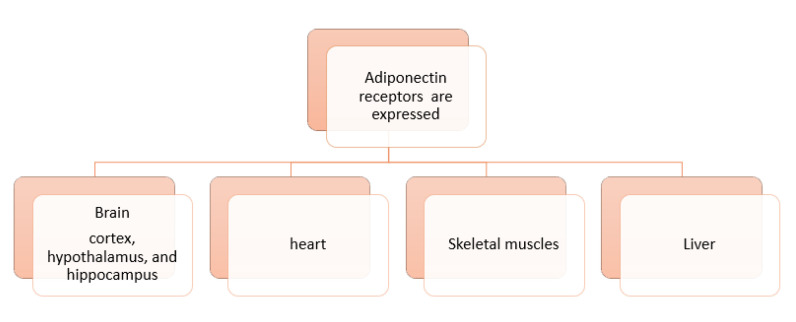
Expression of adiponectin receptors in various organs‘ cells.

**Figure 6 pharmaceutics-14-01430-f006:**
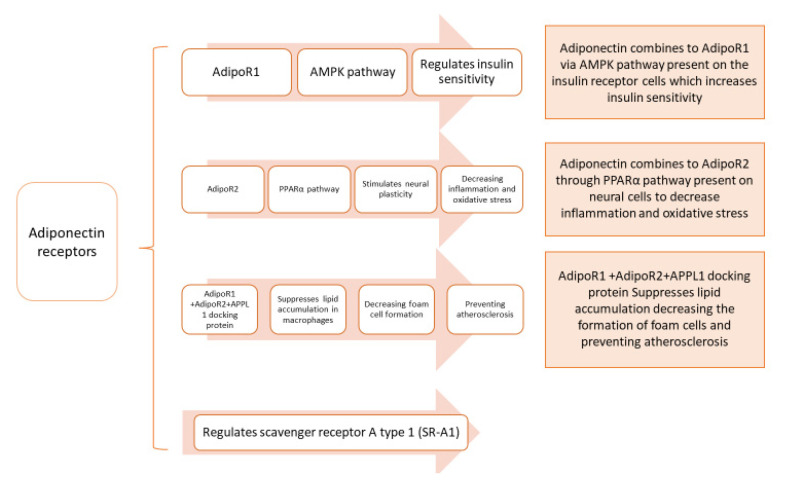
Multiple adiponectin receptor pathways in the cells.

**Figure 7 pharmaceutics-14-01430-f007:**
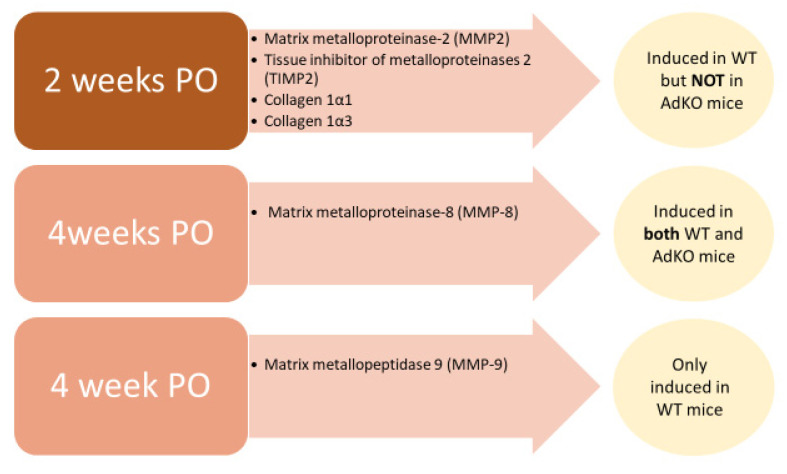
Gene array analysis observation of pressure overload (PO) and duration on adiponectin knockout (AdKO) and wild-type (WT) mice of cardiac cells.

**Table 1 pharmaceutics-14-01430-t001:** Effect of adiponectin mimetic compound on epididymal and subcutaneous inguinal (Sc Ing) adipocytes. The table shows the effect of adiponectin mimetic compound ALY688 on epididymal (Epid) and subcutaneous inguinal (Sc Ing) adipocytes [75].

ALY688 Affect	Epid Adipocytes	Sc Ing Adipocytes
alter basal lipolysis	Was not affect	Was not affected
enhanced isoproterenol-induced lipolysis	affected	-
expression of AdipoR2 mRNA	significantly higher expression	Less expression
AKTSer473 and p38 mitogen-activated protein kinase (MAPK) phosphorylation	Was not affected	Was not affected
Basal and insulin-stimulated rates of glucose uptake	Was not affected	Was not affected
glucose incorporation into lipids	Was not affected	Was not affected

**Table 2 pharmaceutics-14-01430-t002:** Adiponectin in clinical research (adults and children) and in vivo studies.

Research Domain	Adiponectin Effect on Different Health Conditions	Reference
Clinical research and experimental animals	Decrease concentrations of adiponectin are linked with a rise in Risk of obesity Insulin resistance Decreased HDL Increase in TG leading to initiation of vascular disorders	[59]
Clinical research	Decrease of adiponectin concentration in patients with type2 DM was recorded	[60]
Clinical research	In apolipoprotein E deficiency, adiponectin was recognized to have direct effect decreasing inflammatory responses caused by mechanical injury in vascular endothelium and increasing endothelium protection through AMP kinase activation	[2]
Clinical research	Adiponectin concentration and the increase in BNP increased mortality in patients with stable CAD, whereas the increase in adiponectin was not related to CV death or stroke	[68]
Clinical research (children)	Hypoadiponectinemia may act as a factor in the low-grade systemic chronic inflammatory case linked with obesity in children. low concentrations of adiponectin are linked not only with higher C-reactive protein concentrations, but also with parameters of the metabolic disease which includes lipid profile.	[77]
Clinical research (children)	The connection between obesity and parameters of insulin resistance is reduced in youth of higher adiponectin levels, which made adiponectin a potential mediation goal or risk biomarker	[78]
In Vitro studies	Adiponectin modified mesenchymal stem cells revealed protection on cardiac fibrosis so modulated mesenchymal stem cells may be a new therapeutic for the diabetic cardiomyopathy in the future	[86]
In Vitro studies	Mesenchymal stem cells have been used therapeutically in mice for pressure overload-led heart failure involved the existence of plasma adiponectin and T-cadherin expression	[73]

## Data Availability

All information has been provided in this review article.

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
