# Peer review of "Adiponectin System (Rescue Hormone): The Missing Link between Metabolic and Cardiovascular Diseases"

_pharmaceutics, 2022, doi:10.3390/pharmaceutics14071430_

Round 1
Reviewer 1 Report
In this manuscript, they reviewed the physiology and pathophysiology role of the adiponectin system to connect metabolism and cardiovascular diseases. In the cardiovascular system, they focused on the adiponectin plays as a biomarker and potential therapeutic tool. They reviewed the recent articles and represented them in an accurate and objective view. Overall, this article is worth publishing.
Author Response
Thank you for your kind words

Reviewer 2 Report
The review on adiponectin (ADPN) system as rescue hormone is properly and well organized, focused on main functions of this molecule. Adiponectin is presented with respect of the structure, origin, sources and physiological and pathophysiological role. The secretion of ADPN by adipose tissue, endothelial cells and cardiomyocytes is presented along with protective functions against cardiovascular diseases, glucose and insulin control, obesity. Authors underline the promising clinical importance of ADPN as a biomarker of metabolic and cardiovascular syndromes. Very special and new in the literature is a chapter describing receptors of ADPN, their expression and pathways. This "adiponectin system" represents the "hormone-receptor system" well known in endocrinology and leading to specific cellular responses specific target cells. Figure 5 (page 9) shows ADPN receptors pathway, however the final steps of them are presented in very general way. I would suggest to indicate what are the final biochemical effects of those four pathways, for example the AMPK pathway in regulation of insulin sensitivity, or in prevention of atherosclerosis, decreasing inflammation or the oxidative stress. How the receptors or "adiponectin system" participate in those pathways from molecular point of view?
The application of ADPN concentration in blood plasma seems to be very promising biomarker of the progress of atherosclerosis when determined in children first. Is it possible to treat its high level as a risk factor of cardiovascular diseases?
Table 2 sums up the clinical researches and in vitro studies on the effects of ADPN on risk of obesity, diabetes, inflammation, cardiomyopathy, so I found this table as a key characteristic of ADPN level in different health conditions.
The review is concluded in some statements underlying the protective effects of ADPN and its receptors on cardiovascular system.
The choice of references seems to be sufficient and quite recent.
My only remarks are related to minor typing errors, and I am sure they are without influence on the high quality of the article.
Author Response
Reviewer 2 Comments
The review on adiponectin (ADPN) system as rescue hormone is properly and well organized, focused on main functions of this molecule. Adiponectin is presented with respect of the structure, origin, sources and physiological and pathophysiological role. The secretion of ADPN by adipose tissue, endothelial cells and cardiomyocytes is presented along with protective functions against cardiovascular diseases, glucose and insulin control, obesity. Authors underline the promising clinical importance of ADPN as a biomarker of metabolic and cardiovascular syndromes. Very special and new in the literature is a chapter describing receptors of ADPN, their expression and pathways. This "adiponectin system" represents the "hormone-receptor system" well known in endocrinology and leading to specific cellular responses specific target cells. Figure 5 (page 9) shows ADPN receptors pathway, however the final steps of them are presented in very general way. I would suggest to indicate what are the final biochemical effects of those four pathways, for example the AMPK pathway in regulation of insulin sensitivity, or in prevention of atherosclerosis, decreasing inflammation or the oxidative stress. How the receptors or "adiponectin system" participate in those pathways from molecular point of view?
Thank you for your comment. Explanations were added to the figure as shown below (fig.5 now is indicated as fig.6 in the text)
The application of ADPN concentration in blood plasma seems to be very promising biomarker of the progress of atherosclerosis when determined in children first. Is it possible to treat its high level as a risk factor of cardiovascular diseases?
It may not be a risk factor but an indicator, as it ispresent in only certain cases, although a positive link between adiponectin and mortality is present the paradoxal effect is not clear, more studies are required in such these cases.
Table 2 sums up the clinical researches and in vitro studies on the effects of ADPN on risk of obesity, diabetes, inflammation, cardiomyopathy, so I found this table as a key characteristic of ADPN level in different health conditions.
Thank you for your kind remark
The review is concluded in some statements underlying the protective effects of ADPN and its receptors on cardiovascular system.
Thank you for your remark
The choice of references seems to be sufficient and quite recent.
Thank you for your comment
My only remarks are related to minor typing errors, and I am sure they are without influence on the high quality of the article.
Thank you, revisions have been made.

Reviewer 3 Report
The manuscript described the physiological roles of adiponectin as anti-inflammatory and cardioprotective hormone as well as how it plays as a biomarker and a potential therapeutic tool in cardiovascular system and metabolic disease related system. The paradox role of adiponectin relevant to cardiovascular mortality is a worthy of being taken into considerations in the future.
1. There were some wording issues e.g. Ssignificant (Page 12) and several grammar issues, to make it harder to read.
2. A structure figure and the pathway of adiponectin should be considered in ‘Introduction’ Section to facilitate the readers.
3. Figure 1 and 2 could be combined and described more directly with some signal pathways and/or with cellular locations.
4. A figure for the paradox effect would be nice to facilitate the readers.
5. Some descriptions in Figure 3 have been lacked e.g., CTRP9 (on the right side).
6. A summary table of the descriptions in Figure 4 would be nice to have as well as Figure 5.
7. The clinical trial numbers should be cited into Table 2.
8. The whole logical structure of the manuscript is a little bit confusing as it jumps between the metabolic disease and cardiovascular disease before fully describe the potential connection. There are several reviews [1-3] already mentioned the link between the metabolic and cardiovascular diseases. Therefore, a more detailed and logical described manuscript would be expected. How about the manuscript starts with the description of adiponectin, its receptors and reveal its related functions (in vitro, in vivo, clinical aspects) in metabolic diseases, and cardiovascular diseases separately Then, the relationship between the two types of disease with adiponectin and/or its paralogs. Last, the discussions and potential clinical applications of Adiponectin and its receptors.
Reference
[1] Recinella, L., Orlando, G., Ferrante, C., Chiavaroli, A., Brunetti, L., & Leone, S. (2020). Adipokines: new potential therapeutic target for obesity and metabolic, rheumatic, and cardiovascular diseases. Frontiers in physiology, 11, 578966.
[2] Khoramipour, K., Chamari, K., Hekmatikar, A. A., Ziyaiyan, A., Taherkhani, S., Elguindy, N. M., & Bragazzi, N. L. (2021). Adiponectin: Structure, physiological functions, role in diseases, and effects of nutrition. Nutrients, 13(4), 1180.
[3] Field, B. C., Gordillo, R., & Scherer, P. E. (2020). The role of ceramides in diabetes and cardiovascular disease regulation of ceramides by adipokines. Frontiers in endocrinology, 11, 569250.
Author Response
Reviewer 3 Comments
The manuscript described the physiological roles of adiponectin as anti-inflammatory and cardioprotective hormone as well as how it plays as a biomarker and a potential therapeutic tool in cardiovascular system and metabolic disease related system. The paradox role of adiponectin relevant to cardiovascular mortality is a worthy of being taken into considerations in the future.
- There were some wording issues e.g. significant (Page 12) and several grammar issues, to make it harder to read.
Thank you, the paper has been revised.
- A structure figure and the pathway of adiponectin should be considered in ‘Introduction’ Section to facilitate the readers.
Thank you for your comment, a more detailed explanation of the pathways was added to figure 5 (now indicated as fig.6).
- Figure 1 and 2 could be combined and described more directly with some signal pathways and/or with cellular locations.
Thank you for your comment, we wanted the information in both figure 1 and 2 to be simple and easily received by the readers, as figure 1 combines the sources of adiponectin and their effects, while figure 2 contains the Physiological and pathological links of adiponectin to diseases.
- A figure for the paradox effect would be nice to facilitate the readers.
Thank you for your comment, the figure below has been created and added
- Some descriptions in Figure 3 have been lacked e.g., CTRP9 (on the right side).
Thank you for your comment, the CTRP9 on the right side of the figure is indicated as decreased (arrow pointing down) in the plasma.
- A summary table of the descriptions in Figure 4 would be nice to have as well as Figure 5.
Thank you for your kind comment.
We believe that the content in both figures 4 and 5 are simplified and clear, and addition of tables with the same information may be repetitive.
- The clinical trial numbers should be cited into Table 2.
Thank you for your comment, the numbers have been cited in the table and in the text as well.
- The whole logical structure of the manuscript is a little bit confusing as it jumps between the metabolic disease and cardiovascular disease before fully describe the potential connection. There are several reviews [1-3] already mentioned the link between the metabolic and cardiovascular diseases. Therefore, a more detailed and logical described manuscript would be expected. How about the manuscript starts with the description of adiponectin, its receptors and reveal its related functions (in vitro, in vivo, clinical aspects) in metabolic diseases, and cardiovascular diseases separately Then, the relationship between the two types of disease with adiponectin and/or its paralogs. Last, the discussions and potential clinical applications of Adiponectin and its receptors.
Reference
[1] Recinella, L., Orlando, G., Ferrante, C., Chiavaroli, A., Brunetti, L., & Leone, S. (2020). Adipokines: new potential therapeutic target for obesity and metabolic, rheumatic, and cardiovascular diseases. Frontiers in physiology, 11, 578966.
[2] Khoramipour, K., Chamari, K., Hekmatikar, A. A., Ziyaiyan, A., Taherkhani, S., Elguindy, N. M., & Bragazzi, N. L. (2021). Adiponectin: Structure, physiological functions, role in diseases, and effects of nutrition. Nutrients, 13(4), 1180.
[3] Field, B. C., Gordillo, R., & Scherer, P. E. (2020). The role of ceramides in diabetes and cardiovascular disease regulation of ceramides by adipokines. Frontiers in endocrinology, 11, 569250.
Thank you so much for your carful and detailed comment,
The references you kindly mentioned do differ from our review: ref 1: links many adipokines to cardiovascular diseases, while ref 2: links adiponectin to diseases in general, ref 3: concentrates more on ceramides and their regulation by adiponectin.
The main concentration of our review is adiponectin system and the effect on cardiovascular disease, we started with the physiology of adiponectin and its effects on cardiovascular system. In the final version, we explained the paradoxal effects reported on adiponectin, after which we listed the resources and functions of adiponectin, its receptors and the adiponectin system, the effects of adiponectin on neurophysiology of the heart, followed by the physiological link with cardiovascular disease in adults and children, ending with cardiac remodeling.

Reviewer 4 Report
This is a good effort to understand the very complex roles of adiponectin in cell and molecular functions. It is a thorough review of existing literature. The manuscript suffers from many English grammar and composition errors. There are many comma splices, a number misplaced adjectives and places where short sections belong in another part of the paper. For example, the last 5 lines in the introduction would be better in the next section (the cardiovascular section). The reference list needs a good review for consistency of format. All in all, this a good paper that needs a reworking by someone proficient in English.
Author Response
Thank you for your comment, the paper has been revised.
Round 2
Reviewer 3 Report
The multiple signaling pathways of the adiponectin coupled with its cellular locations should be addressed into the manuscript. And the adiponectin structural information should be provided.
Author Response
The multiple signaling pathways of the adiponectin coupled with its cellular locations should be addressed into the manuscript.
Thank you for your valuable comment
This has been added in pages 8 and 10

Reviewer 4 Report
The paper is much improved. There are a few remaining awkward sentences and phrases that can be corrected by the authors or the editorial staff.
Author Response
The paper is much improved. There are a few remaining awkward sentences and phrases that can be corrected by the authors or the editorial staff.
Thank you for your comment
The paper has been revised again.

Round 3
Reviewer 3 Report
No further comments.